# Generalization Performance of Quantum Metric Learning Classifiers

**DOI:** 10.3390/biom12111576

**Published:** 2022-10-27

**Authors:** Jonathan Kim, Stefan Bekiranov

**Affiliations:** 1GSK R&D Stevenage, GlaxoSmithKline, Stevenage SG1 2NY, UK; 2Department of Biochemistry and Molecular Genetics, University of Virginia, Charlottesville, VA 22908, USA

**Keywords:** quantum machine learning, quantum metric learning, kernel method, kernel classifiers

## Abstract

Quantum computing holds great promise for a number of fields including biology and medicine. A major application in which quantum computers could yield advantage is machine learning, especially kernel-based approaches. A recent method termed quantum metric learning, in which a quantum embedding which maximally separates data into classes is learned, was able to perfectly separate ant and bee image training data. The separation is achieved with an intrinsically quantum objective function and the overall approach was shown to work naturally as a hybrid classical-quantum computation enabling embedding of high dimensional feature data into a small number of qubits. However, the ability of the trained classifier to predict test sample data was never assessed. We assessed the performance of quantum metric learning on test ants and bees image data as well as breast cancer clinical data. We applied the original approach as well as variants in which we performed principal component analysis (PCA) on the feature data to reduce its dimensionality for quantum embedding, thereby limiting the number of model parameters. If the degree of dimensionality reduction was limited and the number of model parameters was constrained to be far less than the number of training samples, we found that quantum metric learning was able to accurately classify test data.

## 1. Introduction

Significant progress has recently been made toward the development of fault tolerant quantum computers (FTQCs) [1]. Their development would result in the speedup of many algorithms that are approaching severe limits on classical computers. The range of applications include quantum chemistry [2], search [3], cryptography [4] and machine learning [5]. These applications are relevant to many domains of study including biology and medicine. In the field of machine learning, exponential speedups on a quantum compared to classical computer have been proven [5] for implementing quantum support vector machines [6], quantum Boltzmann machines [7,8], least squares fitting [9], and quantum principal component analysis [10]. Quadratic speedups have been demonstrated [5] for classical Boltzmann machines [11], quantum reinforcement learning [12], online perceptron [13], and Bayesian inference [14,15]. However, these speedups assume a FTQC with high connectivity and hundreds to thousands, even millions for some applications, of qubits. In addition, some of these quantum algorithms require quantum RAM (qRAM) which executes a quantum coherent mapping of a classical vector into a quantum state [16,17], for their quantum advantage over classical computers. However, qRAM hardware has not been developed. Currently, quantum computing is in its noisy intermediate-scale quantum (NISQ) era [18].

A major application in which even NISQ-era quantum computers could yield advantage is kernel-based machine learning [19,20,21,22]. Broadly, two sets of approaches have recently been explored [20,21]: (1) map a large feature space into a quantum state and calculate a kernel function on a quantum computer and make use of this kernel in a classical classifier (e.g., SVM) and (2) apply a variational quantum circuit to classify data on the quantum computer in Hilbert space. Kernel-based classifiers that interfere the test and train data and effectively calculate their Euclidean distance [19,23,24] and/or inner product [23,24,25] have been developed and assessed on IBM quantum computers and performed close to theoretical expectations if the number of gates were kept to a relatively small number [25]. Formally, supervised quantum models have been shown to be kernel methods [22], and it has been suggested that quantum computers could enable kernel-based machine learning in a similar way that GPU-accelerated hardware enabled deep learning [22]. As a result of these developments, a number of kernel-based quantum machine learning studies have been performed in which the trainability [26,27,28,29], expressivity [30,31], robustness [32,33] and generalizability [30,31,33,34,35] of quantum kernel-based models implemented on NISQ-era quantum computers have been studied as well as the extent to which quantum errors can be mitigated on a classical computer [30,31].

In this work, we focus on a quantum kernel-based machine learning approach termed quantum metric learning (QML) [26]. Here, a quantum embedding is learned by maximizing the Hilbert-Schmidt distance of data samples from two classes in such a way that two classes are separated in Hilbert space. This enables a simple linear decision boundary to be implemented in Hilbert space which represents a complex decision boundary in the original feature space. This approach has all the advantages that come with kernel-based approaches mentioned above along with a number of other attractive features for NISQ-era quantum computing including: (1) simple, quantum-based cost function based on the Hilbert-Schmidt distance, (2) seamless applicability as a hybrid quantum-classical approach that reduces the dimensionality of the input feature space for quantum embedding to a small number of qubits, (3) ability to directly visualize the extent to which samples with different class labels are separated and (4) ability to be implemented on a quantum computer as a classifier using multiple swap gates [23,24,25,26]. Despite these highly promising attributes of QML, the primary manuscript detailing the method [26] only demonstrated its ability to separate training data. The ability of QML to generalize well by assessing a trained model on test data was not shown. Consequently, we fill this gap by training and testing QML with the original ImageNet Hymenoptera Dataset containing images of ants and bees [36] as well as the University of California Irvine Machine Learning Breast Cancer Wisconsin (Diagnostic) dataset [37]. The breast cancer dataset contains 30 normalized clinical features for each breast cancer patient whose tumor was diagnosed as malignant and benign. We used precision, recall and F1-score as performance metrics for test data. We also report the resulting cost function for both train and test data. We reproduced the result that for the original ant and bee image data, we were able to achieve a high level of separation on training data. However, we found that the trained classifier did not perform well on hold out test data. We noticed that the number of model parameters exceeded the number of training samples, so we hypothesized that the model was overfitting the training data. Application of principal component analysis (PCA) to reduce the input feature dimension and number of model parameters did not significantly improve test performance on this dataset. We turned to the breast cancer data which contained far fewer input features and more samples and further applied PCA as well to reduce the input feature dimensions and number of model parameters. We found that QML was able to perform well on both training and test data in this setting. Thus, when adhering to conventional bias-variance principles, namely, constraining the number of model parameters to be notably less than the number of training samples, we find that QML-based classifiers generalize well. This is true as long as the initial number of features (i.e., the number of features prior to PCA) is not too high.

## 2. Materials and Methods

### 2.1. Quantum Metric Learning Expressed as a Kernel-Based Quantum Model

In quantum metric learning, a quantum embedding,
(1)|x〉=Φ(x,θ)|0…0〉,
is learned where Φ(x,θ) is a feature map which maps the input data *x* to a quantum state |x〉 which separates the data according to class labels in Hilbert space by maximizing the Hilbert-Schmidt distance $D_{hs}$ or, equivalently, by minimizing a cost function *C* defined in terms of $D_{hs}$, through gradient descent of the model parameters θ. The Hilbert-Schmidt distance is
(2)$$D_{hs}\left(\rho ,\sigma \right)=\mathrm{tr}\left[{\left(\rho -\sigma \right)}^{2}\right],$$
where ρ and σ are density matrices representing ensembles of $M_{a}$ and $M_{b}$ training data points *a* and *b* from class *A* and *B*, respectively: (3)$$\rho =\frac{1}{M_{a}}\sum_{a\in A}|a\rangle \langle a|$$
and
(4)$$\sigma =\frac{1}{M_{b}}\sum_{b\in B}|b\rangle \langle b|.$$

The cost function *C*, whose range is [0,1], that is minimized is
(5)$$C=1-\frac{1}{2}D_{hs}\left(\rho ,\sigma \right).$$

Once *C* is minimized, the parameters, θ, of the feature map are determined in such a way that the training data {a,b} is separated in Hilbert space. In order to classify a test sample, *x*, it must first be embedded using the feature map as shown in Equation (Equation 1). A fidelity classifier [23,24,25,26] can then be defined by the difference in squared inner product between the embedded test sample |x〉 and the respective class *A* and *B* embedded training samples {|a〉,|b〉}: (6)$$\begin{array}{cc} f(x) & =\frac{1}{M_{a}}\sum_{a\in A}{\left|\langle a|x\rangle \right|}^{2}-\frac{1}{M_{b}}\sum_{b\in B}{\left|\langle b|x\rangle \right|}^{2} \end{array}$$(7)$$\begin{array}{cc} \; & =\langle x|\rho -\sigma |x\rangle . \end{array}$$

Equation (7) can be viewed as an expectation of a measurement, M, where
(8)$$\begin{array}{cc} M & =\rho -\sigma \end{array}$$
(9)$$\begin{array}{cc} \; & =\frac{1}{M_{a}}\sum_{a\in A}|a\rangle \langle a|-\frac{1}{M_{b}}\sum_{b\in B}|b\rangle \langle b|. \end{array}$$

Thus, the fidelity classifier may be expressed as follows: (10)$$\begin{array}{cc} f(x) & =\langle x|M|x\rangle \end{array}$$(11)$$\begin{array}{cc} \; & =\mathrm{tr}[|x\rangle \langle x|M]. \end{array}$$

Equation (11) is the definition of a quantum model (see Equation (34) of Schuld et al. [22]) which can be expressed as a quantum kernel-based model. We implement and assess the generalization performance of quantum metric learning using the following k-nearest neighbor (KNN) kernel-based classifier: (12)$$\begin{array}{cc} \hat{y} & =\mathrm{sgn}(f(x)) \end{array}$$(13)$$\begin{array}{cc} \; & =\mathrm{sgn}(\sum_{a\in A}\alpha_{a}\kappa \left(a,x\right)-\sum_{b\in B}\alpha_{b}\kappa \left(b,x\right)), \end{array}$$
where $\hat{y}$ is the prediction for test sample *x* and sgn denotes the sign function. Comparison of Equations (Equation 6) and (Equation 13) yields the result that $\alpha_{a}=1/M_{a}$, $\alpha_{b}=1/M_{b}$,
(14)$$\kappa \left(a,x\right)={\left|\langle a|x\rangle \right|}^{2},$$
and
(15)$$\kappa \left(b,x\right)={\left|\langle b|x\rangle \right|}^{2},$$
where κ(a,x) and κ(b,x) are defined as quantum kernels (see Equation (6) of Schuld et al. [22]) which are the inner product between the embedded test data, *x*, and training data, *a* and *b*, respectively, in the context of a KNN classifier.

### 2.2. The Quantum Metric Learning Embedding Circuit

Various adaptations of Lloyd et al.’s hybrid quantum metric learning embedding [26] were used throughout this work. See Figure 1A for a full illustration of the general embedding. The quantum component of the algorithm (the trainable *quantum feature map*, a repeating circuit ansatz consisting of single-qubit $R_{x}$, $R_{y}$ rotation gates and two-qubit ZZ coupling gates [26,38] resulting in 12 trainable quantum parameters) was left unchanged. The classical components leading to the intermediate $x_{1}$ and $x_{2}$ inputs to the quantum feature map were replaced and varied. We note that the quantum circuit is precisely the same as that of Lloyd et al. [26]. The example ansatz in Figure 3 of [26] is for three inputs ($x_{1}$, $x_{2}$ and $x_{3}$). However, we and Lloyd et al. [26] use two inputs ($x_{1}$ and $x_{2}$) to assess QML on real world datasets.

We now describe the effects that the $R_{x}\left(x_{i}\right)$, $R_{y}\left(\theta_{j}\right)$ and $\mathrm{ZZ}(\theta_{j})$ gates have on the two-qubit state at the *k*th stage of the circuit, $|x_{k}\rangle$, where i=1,2, j=1,2,...,12, k=1,2,...,14 and (16)$$|x_{k}\rangle =\alpha_{k}|00\rangle +\beta_{k}|01\rangle +\gamma_{k}|10\rangle +\delta_{k}|11\rangle .$$ For example, $|x_{1}\rangle$ = |00〉 with $\alpha_{1}=1$ and $\beta_{1}=\gamma_{1}=\gamma_{1}=0$ is the initial two-qubit state entering the circuit on the left of Figure 1A. The state $|x_{14}\rangle =|x\rangle$ is the final state shown on the right of the circuit in Figure 1A. The operation of the first $R_{x}\left(x_{1}\right)$ and $R_{x}\left(x_{2}\right)$ gates yields $|x_{2}\rangle$, where (17)$$\begin{array}{cc} \alpha_{2} & =\frac{1}{2}\cos\left(\frac{x_{1}+x_{2}}{2}\right)+\frac{1}{2}\cos\left(\frac{x_{1}-x_{2}}{2}\right), \end{array}$$(18)$$\begin{array}{cc} \beta_{2} & =-\frac{i}{2}\sin\left(\frac{x_{1}+x_{2}}{2}\right)+\frac{i}{2}\sin\left(\frac{x_{1}-x_{2}}{2}\right), \end{array}$$(19)$$\begin{array}{cc} \gamma_{2} & =-\frac{i}{2}\sin\left(\frac{x_{1}+x_{2}}{2}\right)-\frac{i}{2}\sin\left(\frac{x_{1}-x_{2}}{2}\right) \end{array}$$ and (20)$$\delta_{2}=-\frac{1}{2}\cos\left(\frac{x_{1}-x_{2}}{2}\right)+\frac{1}{2}\cos\left(\frac{x_{1}+x_{2}}{2}\right).$$

We see that the two-qubit state becomes angularly embedded by a combination real and complex coefficients containing sine and cosine functions. The operation of the first $\mathrm{ZZ}(\theta_{1})$ entangler gate yields $|x_{3}\rangle$, where (21)$$\begin{array}{cc} \alpha_{3} & =e^{\frac{-i\theta_{1}}{2}}\alpha_{2}, \end{array}$$(22)$$\begin{array}{cc} \beta_{3} & =e^{\frac{i\theta_{1}}{2}}\beta_{2}, \end{array}$$(23)$$\begin{array}{cc} \gamma_{3} & =e^{\frac{i\theta_{1}}{2}}\gamma_{2} \end{array}$$ and (24)$$\delta_{3}=e^{\frac{-i\theta_{1}}{2}}\delta_{2}.$$ The operation of the $R_{y}\left(\theta_{2}\right)$ and $R_{y}\left(\theta_{3}\right)$ gates then yields $|x_{4}\rangle$, where (25)$$\begin{array}{cc} \alpha_{4} & =\frac{\alpha_{3}-\delta_{3}}{2}\cos\left(\frac{\theta_{2}+\theta_{3}}{2}\right)+\frac{\alpha_{3}+\delta_{3}}{2}\cos\left(\frac{\theta_{2}-\theta_{3}}{2}\right)\\ & -\frac{\beta_{3}+\gamma_{3}}{2}\sin\left(\frac{\theta_{2}+\theta_{3}}{2}\right)+\frac{\beta_{3}-\gamma_{3}}{2}\sin\left(\frac{\theta_{2}-\theta_{3}}{2}\right), \end{array}$$
(26)$$\begin{array}{cc} \beta_{4} & =\frac{\beta_{3}+\gamma_{3}}{2}\cos\left(\frac{\theta_{2}+\theta_{3}}{2}\right)+\frac{\beta_{3}-\gamma_{3}}{2}\cos\left(\frac{\theta_{2}-\theta_{3}}{2}\right)\\ & +\frac{\alpha_{3}-\delta_{3}}{2}\sin\left(\frac{\theta_{2}+\theta_{3}}{2}\right)-\frac{\alpha_{3}+\delta_{3}}{2}\sin\left(\frac{\theta_{2}-\theta_{3}}{2}\right), \end{array}$$
(27)$$\begin{array}{cc} \gamma_{4} & =\frac{\beta_{3}+\gamma_{3}}{2}\cos\left(\frac{\theta_{2}+\theta_{3}}{2}\right)-\frac{\beta_{3}-\gamma_{3}}{2}\cos\left(\frac{\theta_{2}-\theta_{3}}{2}\right)\\ & +\frac{\alpha_{3}-\delta_{3}}{2}\sin\left(\frac{\theta_{2}+\theta_{3}}{2}\right)+\frac{\alpha_{3}+\delta_{3}}{2}\sin\left(\frac{\theta_{2}-\theta_{3}}{2}\right) \end{array}$$ and (28)$$\begin{array}{cc} \delta_{4} & =-\frac{\alpha_{3}-\delta_{3}}{2}\cos\left(\frac{\theta_{2}+\theta_{3}}{2}\right)+\frac{\alpha_{3}+\delta_{3}}{2}\cos\left(\frac{\theta_{2}-\theta_{3}}{2}\right)\\ & +\frac{\beta_{3}+\gamma_{3}}{2}\sin\left(\frac{\theta_{2}+\theta_{3}}{2}\right)+\frac{\beta_{3}-\gamma_{3}}{2}\sin\left(\frac{\theta_{2}-\theta_{3}}{2}\right). \end{array}$$

In this way, we see that we get growing products of sine and cosine components (in terms of both the linear trainable parameters, $x_{i}$, and the ’quantum’ trainable parameters, $\theta_{j}$) in each element of the resulting vector as we progress through the circuit. As the circuit ansatz is repeated further, this results in an increase in both the ’sharpness’ and the number of peaks and troughs representing the angular embedded data, allowing for the high levels of expressivity needed for effective embedding.

When working with the Hymenoptera ants and bees image dataset, the replaceable classical part of the embedding consisted of images of ants and bees that had been standardized and normalized. We explored passing them through a pre-trained ResNet-18 network (without the final layer) as well as working with them directly. The first approach resulted in 512 classical input features [26,39], while the second approach yielded 150528 classical input features. In the second approach, the features were then always dimensionally reduced via PCA to prevent there being an exceptionally high number of trainable parameters. When working with the breast cancer dataset, the replaceable classical part of the embedding corresponded to 30 normalized input clinical features. This resulted in 30 classical input features.

These *n* classical input features were then multiplied by a 2×n parameter matrix, resulting in 2n trainable linear parameters and the two inputs ($x_{1}$, $x_{2}$) to the quantum feature map. In many cases in both datasets, the initial input features also underwent dimensional reduction through principal component analysis (PCA) to yield lower values of *n*, so as to help minimize overfitting by the subsequent models. However, when working with this dimensional reduction approach, it was also important not to reduce the number of linear parameters too much so as to retain the expressivity of the models.

### 2.3. Training the Quantum Metric Learning Models

The quantum feature map itself provided 4×3=12 trainable quantum parameters (i.e., 4 repeated circuit ansatzes containing 3 parameters per ansatz) and as such, every model consisted of 2n+12 total trainable parameters. Each model was randomly initialized and trained for 1500 steps with a batch size of 10, using the root mean squared propagation (RMSProp) optimizer with a step size of 0.01. With successful training, each new ($x_{1}$, $x_{2}$) input to the model becomes embedded into a state |x〉 in Hilbert space such that the Hilbert-Schmidt distances between the embedded states of opposing classes, shown in Equation (Equation 2), are maximized or equivalently, the Hilbert-Schmidt cost function, Equation (Equation 5), is minimized. The hybrid parameter optimization steps were performed using the PennyLane software package [40] and the embedded data were subsequently classified by a k-nearest neighbor (KNN) classifier.

### 2.4. ImageNet Hymenoptera Dataset

The first dataset used to explore Lloyd et al.’s quantum metric learning embedding [26] was the ImageNet Hymenoptera image dataset [36]. This dataset consists of 397 colored images of ants and bees in various environments. Each sample can thus be assigned a class of either *ant* or *bee*. By default, the dataset is split into a training set and a test set in the approximate ratio of 3:2. This train-test split was manually changed at times, as dictated by a random seed. Each image was standardized into a resolution of 224×224 then normalized using the PyTorch Normalize function [41] to yield ImageNet’s preferred mean pixel values of (0.485, 0.456, 0.406) and standard deviation pixel values of (0.229, 0.224, 0.225) [36,41]. Notably, this ants/bees dataset is the same dataset as the one used by Lloyd et al. in their paper [26], as well as by Mari et al. in their 2019 paper on *quantum transfer learning* [39].

#### 2.4.1. Training QML Models with Feature Extraction Using ResNet-18

The first step in assessing the hybrid embedding was to investigate the resulting training cost, test cost, test set precision, test set recall and test set F1-score using the same embedding setup as presented in the demo code associated with Lloyd et al.’s paper [26]. This setup includes the pre-trained ResNet-18 component which converts each normalized ant or bee image into 512 input features. The 2×512=1024 resulting linear parameters and 12 quantum parameters of the quantum feature map were optimized as detailed above.

#### 2.4.2. Training QML Models with Feature Extraction Using ResNet-18 Followed by PCA

To help address the potential issue of overfitting due to the high number of parameters used when training the linear half of the model, principal component analysis (PCA) was performed on the 512 output features of the ResNet. First, for each sample, the ResNet output features were normalized using the scikit-learn StandardScaler function [42], resulting in a mean of 0 and a standard deviation of 1 for each feature. The 512 normalized features were then reduced to 256, 64, 16, 4 and 2 principal components, leading also to a reduction in the number of linear model parameters. For instance, whenever the features of each sample were reduced to 256 principal components, the model would be trained with 512 linear parameters (as opposed to the original 1024 linear parameters). With a reduction to 4 principal components, the model would be trained using just 8 linear parameters. In general, *n* principal components were multiplied by a 2×n matrix to yield the two-dimensional ($x_{1}$, $x_{2}$) values used as input to the quantum feature map. The elements of the 2×n matrix change between each training iteration, acting as 2n trainable linear parameters. Other than this change to the number of trainable parameters, the training setup of the optimizer was kept the same as in the non-PCA case.

#### 2.4.3. Training QML Models with Feature Extraction Using PCA

A more direct form of PCA was also used on the Hymenoptera image dataset. Instead of passing the images through a ResNet first, the 224×224×3=150528 normalized pixel datapoints per image were reduced directly down to 256, 64, 16, 4 and 2 principal components. As before, these principal components were then multiplied by a 2×n parameter matrix to yield the ($x_{1}$, $x_{2}$) values and 2n trainable linear parameters, where *n* is the number of principal components. These ($x_{1}$, $x_{2}$) values were again used as input to the quantum feature map, then optimized using the optimization approach detailed above.

### 2.5. UCI ML Breast Cancer Wisconsin (Diagnostic) Dataset

The second dataset that we used was the University of California Irvine Machine Learning Breast Cancer Wisconsin (Diagnostic) Dataset [37]. This dataset consists of 569 breast cancer samples, each associated with 30 quantitative values such as cell radius, symmetry and smoothness. Each sample in the dataset can be classified as either *benign* or *malignant*. At different points in this work, the dataset was manually divided into different train-test splits (as determined by set random seeds), each in the ratio of 3:2.

#### 2.5.1. Training QML Models Using All Input Features

As with the Hymenoptera dataset, the goal was to establish how well the hybrid embedding generalizes. To begin, the 30 quantitative attributes of the breast cancer dataset were normalized using the scikit-learn StandardScaler function [42], such that the mean and standard deviation of each attribute became 0 and 1, respectively. The normalized attributes were then matrix-multiplied with a 2×30 parameter matrix, resulting in a set of $x_{1}$ and $x_{2}$ values associated with each sample, as well as a set of 60 trainable linear parameters corresponding to the elements in the matrix. Mirroring the steps that were performed on the Hymenoptera dataset, the 60 linear parameters and 12 quantum parameters were then trained as detailed above. For this dataset, two sets of results were collected in separate tables. Each set of results came from a different pseudo-random train-test split of the data as determined by a random seed. Two sets of results were obtained to account for potential bias in the splits caused by chance.

#### 2.5.2. Training QML Models with Feature Extraction Using PCA

Taking the same approach as with the Hymenoptera dataset, PCA was also performed on the 30 normalized features of the breast cancer dataset to reduce the number of trainable parameters.

Two new sets of models were trained according to the same train-test splits as established in the non-PCA case. Each of these two sets consisted of models trained from 30, 16, 8, 4 and 2 principal components. Just as with the Hymenoptera dataset, the resulting principal components were multiplied by a 2×n parameter matrix where *n* is the number of principal components. This approach yields ($x_{1}$, $x_{2}$) values and 2n linear parameters needed for training and embedding. The same optimizer configuration was used as in all prior cases.

### 2.6. Assessing Quantum Metric Learning Model Performance

For all the QML models generated for both datasets, training costs, test costs, test set precision scores, test set recall scores and test set F1-scores resulting from each of the train-test splits were calculated. $x_{1}$, $x_{2}$ scatter plots and Hilbert space mutual data overlap matrices were generated to examine the level of expressivity of the models and to further review the ability of these models to separate and classify test data.

## 3. Results

### 3.1. Hymenoptera Dataset

As detailed in the Methods section, we trained and tested the Hymenoptera image and Breast Cancer Wisconsin (Diagnostic) datasets using the hybrid classical-quantum classifier shown in Figure 1A. We started with the Hymenoptera dataset using the same approach as Lloyd et al. [26]. In Figure 1B,C, we show a scatter plot of the inputs to the quantum circuit for train and test data before and after 1500 steps of training, respectively. Figure 1B illustrates that we recapitulate the ability of quantum metric learning (QML) to perfectly separate the Hymenoptera image training data when using the ResNet-18 layer with 512 input features in the same way as is seen in Lloyd et al.’s work [26]. With 1024 linear parameters and 12 quantum parameters, the training set ($x_{1}$, $x_{2}$) datapoints seem to cluster very well two-dimensionally after 1500 steps. In contrast, as shown in Figure 1C, the test set datapoints remain very poorly separated. This contrast in separability suggests that the model is severely overfitting in this case.

Figure 2 illustrates the Hilbert space mutual data overlap gram matrices demonstrating the classifiability associated with the training and test results provided in Figure 1C. As expected from a case that shows a high level of overfitting, the training data is separated almost perfectly in Hilbert space (as seen in Figure 2B) while the test data remains barely separated at all (as seen in Figure 2D), demonstrating that the embedding generalizes poorly with the Hymenoptera dataset.

Summarised in Table 1 are the results of training the model on the Hymenoptera dataset in various ways. A specific random seed of ’123’ was used for the train-test split in every row other than the first. The first row uses the same default train-test split as was used in Lloyd et al.’s work [26]. It also corresponds to the results shown in Figure 1 and Figure 2.

Test set F1-score and precision are maximized when using the original setup involving the full 512 output features of ResNet-18 with no further feature reduction through PCA. Training cost is minimized at 512 features, but the corresponding test cost is high, which provides further evidence of overfitting and poor generalization. This also means that the minimized training cost of 0.0141 is likely achieved only when overfitting the training data. The lowest test cost, which is achieved with 256 principal component features and no ResNet step, is hardly reduced from its maximum value of 1. The test set recall is maximized at 256 principal component features with the ResNet step.

Although the 512 feature setup and 256 principal component feature setups seemed to perform slightly better than entirely random class assignment, the resulting scores are still very poor. The highest F1-score being just 0.5912 and the lowest test cost still being as high as 0.9859. Furthermore, regardless of whether or not a ResNet step was used, subsequent feature reduction through PCA only worsened F1-score while drastically increasing training costs. Thus, after reducing the number of parameters, there seems to have been a drop in expressivity, which prevented overfitting. However, this was due to training costs becoming much worse. While it can be said than none of the models in Table 1 demonstrate good test set classification performance, the observed ability for PCA to prevent overfitting is still worth noting, despite it being achieved exclusively through increased training cost values in this case.

### 3.2. Breast Cancer Dataset

Figure 3A,B illustrate the effects of training the hybrid model for 1500 iterations on the breast cancer dataset. Both the training set and test set ($x_{1}$, $x_{2}$) values seem to have separated reasonably well in two dimensions, which contrasts with the Hymenoptera dataset result where only the training set separated well. However, neither set separates well enough for entirely distinct non-overlapping clusters to form (as was seen in Figure 1C). While the training set datapoints in Figure 1 separated into very tight clusters that were isolated from other surrounding clusters, the clusters in Figure 3B are much broader and less well defined. This more modest training set separation, in conjunction with the much greater similarity between the training set clusters and test set clusters indicates that the level of overfitting is much lower when using the breast cancer dataset.

Figure 4 depicts the Hilbert space mutual data overlaps (i.e., ${\left|\langle x|x^{{}^{\prime }}\rangle \right|}^{2}$) associated with the training and test scatter plot results shown in Figure 3. It is clear from Figure 4B,D that both the training set embeddings and the test set embeddings separate relatively well in Hilbert space when using the trained model. The Hilbert space separation and resulting classifiability of the test set appear comparable to those of the training set, which serves as further evidence that overfitting is less of an issue with this dataset.

However, the test set is still classified observably worse than the training set, as seen by the significantly misplaced ’lines’ of overlap present within Figure 4D. Thus, despite the improvements compared to the previous dataset, there is still a moderate level of overfitting occurring. Consequently, there is still room for generalization performance to be improved further.

Figure 5 demonstrates the effects of carrying out PCA on the 30 input features of the breast cancer dataset. As seen in Figure 5B,D, PCA seems to bring both training set and test set ($x_{1}$, $x_{2}$) values into tighter two-dimensional clusters compared to those seen in Figure 3B. This generally has the effect of reducing the relative surface area of the borders between neighboring clusters, which could potentially correlate with better classification after subsequent embedding.

It is worth noting that in Figure 5A (with feature reduction to 8 principal components), the ($x_{1}$, $x_{2}$) values seem to start off reasonably well separated in two-dimensions as a result of the prior PCA step. Then after 1500 steps of training, Figure 5B shows how the model is able to further separate the values such that much more distinctive, globular clusters are formed with a much lower relative surface area where the clusters meet. In contrast, Figure 5C shows that the ($x_{1}$, $x_{2}$) values resulting from 4 principal components begin in a much less well separated two-dimensional state after the initial PCA step. Despite this, the trained model is still able to separate the values into quite distinctive clusters, as shown in Figure 5D. In fact, the two-dimensional area of cluster overlap in Figure 5D still seems to be slightly smaller than the area of cluster overlap in Figure 3B. In other words, regardless of whether PCA is able to group the pre-training ($x_{1}$, $x_{2}$) values by class, the resulting post-training test set is well separated. Interestingly, the PCA-based post-training separation (Figure 5B,D) appears to be better than its non-PCA counterpart (Figure 3B). Thus, we find that feature reduction through PCA can consistently contribute to better generalization performance for this dataset.

A final observation is that the 8 principal component model (with 16 trainable linear parameters) seems to demonstrate greater expressivity than the 4 principal component model (with 8 trainable linear parameters). While the 8 principal component model moves the ($x_{1}$, $x_{2}$) values into more distinctive, globular clusters, the 4 principal component model instead moves the values into a simpler, more linear shape. It seems that having fewer trainable linear parameters can cause the model to lose expressively, leading to less well-defined clusters and perhaps worse post-embedding classification. However, as seen in Figure 1C and Figure 3B, having too many parameters, and, thus, too much expressivity for a limited number of samples, can lead to overfitting and noisier clustering.

Figure 6 illustrates the mutual test data overlaps in Hilbert space (i.e., ${\left|\langle x|x^{{}^{\prime }}\rangle \right|}^{2}$) that correspond to the scatter plots from Figure 5. After training, the purple and yellow tiles seem to have separated better when 8 principal components were used (Figure 6B) compared to when 4 principal components were used (Figure 6D). In particular, there are overall not as many ’lines’ of misassigned overlap running across the four grouped squares in Figure 6B. This suggests that that the 8 principal component model is better at maximally separating embedded test data in Hilbert space than the 4 principal component model and is thus better at classifying new data. This aligns with the higher expressivity observed within the 8 principal component clusters of Figure 5B. Not surprisingly, there appears to be an optimal number of principal components for a given number of samples, which yields the best embedding ability, model expressivity and generalizability.

Summarised in Table 2 and Table 3 are the results of training the hybrid model on the breast cancer dataset in various ways. A different random seed (for creating a pseudo-random pre-determined train-test split) was used for each table. Within each table, the random seed of choice (and thus the specific train-test split) stays consistent. Table 3 also corresponds to the results in Figure 3, Figure 4, Figure 5 and Figure 6.

We emphasize that the differences between the results of Table 2 and Table 3 come solely from the differences in random seeds used. In both result sets, test set F1-score is maximized and test cost is minimized when PCA is performed to produce 8 principal components. Meanwhile, training cost is minimized when all 30 principal components are used (i.e., the same as the initial number of features in the dataset). Test set precision and recall are maximized at either 8 or 16 principal components in each case and are all much higher than the Hymenoptera test set precision and recall scores from Table 1.

Based on our analysis of the breast cancer dataset, it is evident that lowering the number of input features through PCA (thus lowering the number of trainable linear parameters) reduces the level of overfitting by the trained hybrid model. This is observed in the shrinking difference between training costs and test costs. This arises from increases in training costs and is sometimes coupled with decreases in test costs, as well as improvement in test set F1-scores. However, when there are too few linear parameters, F1-scores and test costs worsen again. This is consistent with the observations made relating to Figure 5B,D, where a reduction in the number of features caused the clusters to be more linear (less globular) in shape, pertaining to a decrease in expressivity.

For this particular dataset, reducing the 30 initial features to 8 principal components (16 trainable linear parameters) seems to be the ideal compromise for good generalizability in terms of minimizing overfitting while maximizing expressivity.

## 4. Discussion

We implemented the hybrid classical-quantum machine learning approach termed quantum metric learning [26]. Specifically, we addressed the following gap: while the approach was shown to separate training samples perfectly on a Hymenoptera dataset containing images of ants and bees, the performance of the trained models on hold out test data was not assessed. When using the same circuit, dataset and train-test split as seen in Lloyd et al.’s paper [26], it was found that the resulting hybrid model severely overfits the training data and generalizes poorly. While almost perfect Hilbert space-embedded separation was achieved with the training data, the test data yielded very poor results with an F1-score of only 0.5912. Reducing the number of linear parameters through principal component analysis (PCA) produced even worse outcomes for both the training set and the test set. This is likely due to a decrease in model expressivity. Specifically, a drop in test set recall and F1-score was observed, along with a very steep increase in training cost. The increase in training cost was so dramatic (from 0.0141 to ≥0.9700) that the training cost values became comparable to those of the test. After omitting the ResNet-18 step and carrying out PCA directly on the pixel data, there were no improvements to the results. We found that no method resulted in even modest generalizability for this dataset which had a large number of features compared to the number of samples.

The breast cancer dataset consists of a significantly smaller number of features, while having a greater number of total samples. Even without carrying out PCA, the trained models seemed to generalize reasonably well for the test data, yielding high F1-scores of 0.9396 and 0.9456. However, there was still some evidence of overfitting, with training costs of 0.1727 and 0.2026 being associated with much higher test costs of 0.3623 and 0.2791, respectively. When PCA was performed on the initial features, resulting test set F1-scores were always higher than that of their non-PCA counterpart, while differences between the training costs and test costs were often much lower. Not surprisingly, we also found that test costs and F1-scores tended to worsen again if the number of principal components was too low. For the breast cancer dataset, the ideal balance of high expressivity and low overfitting needed for good generalization was found to be at 8 principal components (16 linear parameters). This yielded an F1-score as high as 0.9722 and a test cost as low as 0.2646 (with a similar training cost of 0.2497). Of course, the optimal number of principal components would vary depending on the dataset.

Quantum metric learning models appear to follow the traditional bias-variance constraints, namely, good generalization results if the number of model parameters is significantly lower than the number of training samples. The above requirements are fulfilled by the breast cancer dataset, where there are 72 initial parameters (resulting from just 30 initial features) and as many as 357 training samples. The initial 72 parameter model generalized well and parameter reduction through PCA served to improve this generalization even further, most notably after a reduction to just 28 model parameters. In contrast, the Hymenoptera dataset has as many as 1036 initial parameters (resulting from at least 512 initial features) while having only 244 training samples; the initial 1036 parameter model generalized poorly and parameter reduction through PCA offered no significant improvement.

For future explorations, it would be insightful to vary the shape of the quantum feature map (and thus the number of quantum parameters involved) and to assess the subsequent effects this has on the expressivity and overfitting observed in any resulting trained models. The quantum feature map can be varied both in its length (the number of ’horizontal’ repetitions of each gate) and its width (the number of qubits used). It could be the case that varying the dimensions of the quantum feature map changes the ideal ratio between the number of initial parameters and the number of samples to achieve good generalization performance. It would also be valuable to explore methods of dimensional reduction other than PCA, such as classical or quantum auto-encoding. Comparisons in generalization performance and classification accuracy between quantum metric learning and other methods of classification (using the breast cancer dataset, as well as a broad range of other datasets) would also be insightful.

## Figures and Tables

**Figure 1 biomolecules-12-01576-f001:**
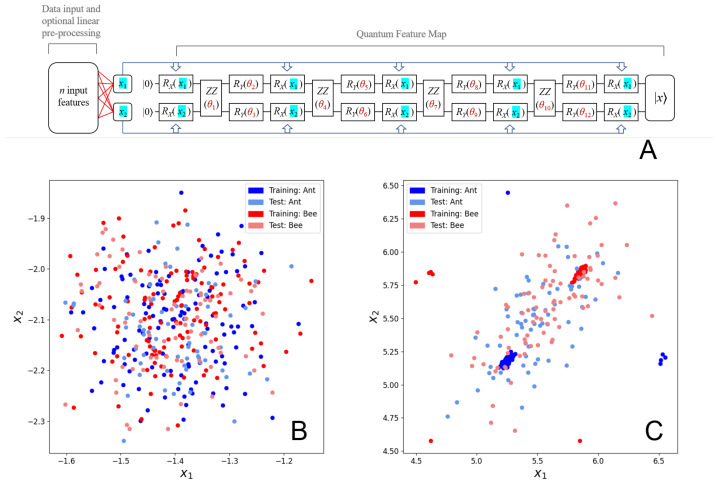
Diagrams illustrating the training process and results of the quantum feature map: (**A**) Diagram of the general quantum metric embedding used. The model takes *n* classical input features and reduces them to two intermediary values ($x_{1}$ and $x_{2}$) through matrix multiplication with a 2×n parameter matrix, whose elements behave as trainable linear parameters. Thus, *n* input features yield 2n trainable linear parameters. The resulting intermediate ($x_{1},x_{2}$) values are then used as input alongside 12 trainable ’quantum’ parameters ($\theta_{1}$–$\theta_{12}$) to progress through the quantum feature map. Each sample ultimately ends up in the embedded |x〉 state in which the Hilbert-Schmidt distance between different classes is maximized through iterative training of the linear and quantum parameters. The illustrated approach represents a generalized adaption of the hybrid quantum metric learning embedding used by Lloyd et al. [26] (**B**) Scatter plot of the ($x_{1},x_{2}$) values of the Hymenoptera dataset with 512 ResNet features (corresponding to 1024 trainable linear parameters) after 0 steps of training. Datapoints from both the training set and the test set are depicted. We note that we used precisely the same train and test samples as in the original study [26] for the Hymenoptera data which corresponded to 61% train and 39% test. (**C**) Scatter plot of the ($x_{1},x_{2}$) values of the Hymenoptera dataset with 512 ResNet features after 1500 steps of training using the PennyLane software package [40]. Datapoints from both the training set and the test set are depicted.

**Figure 2 biomolecules-12-01576-f002:**
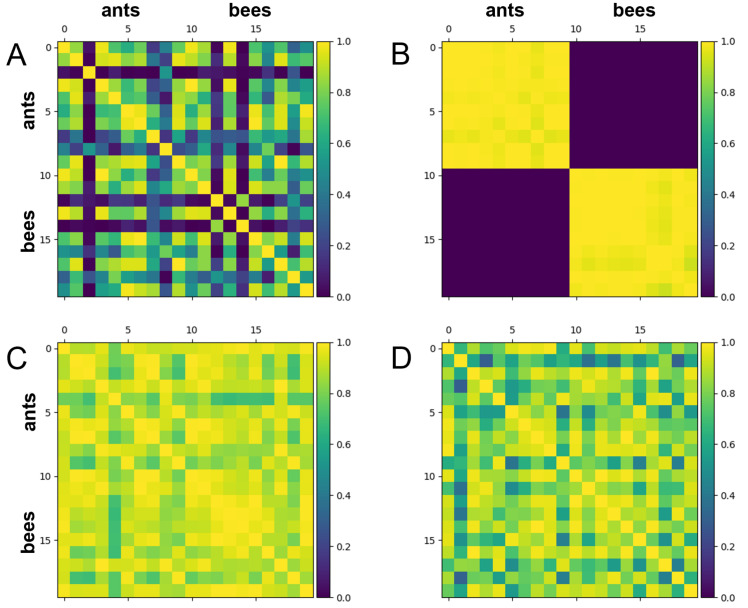
Gram matrices for mutual data overlap (i.e., ${\left|\langle x|x^{{}^{\prime }}\rangle \right|}^{2}$) in Hilbert space for 10 ant and 10 bee samples from the Hymenoptera dataset where 0 and 1 correspond to no and perfect overlap, respectively. In each case, 512 ResNet features (corresponding to 1024 trainable linear parameters) were used. The stronger the separation between the purple tiles (bees) and the yellow tiles (ants), the better the model’s ability to classify. The Hymenoptera dataset’s default train-test split was used for these results. The PennyLane software package was used to train the embedding [40]. (**A**) Mutual data overlap in Hilbert space for training set datapoints at optimization step 0. (**B**) Mutual data overlap in Hilbert space for training set datapoints at optimization step 1500. (**C**) Mutual data overlap in Hilbert space for test set datapoints at optimization step 0. (**D**) Mutual data overlap in Hilbert space for test set datapoints at optimization step 1500.

**Figure 3 biomolecules-12-01576-f003:**
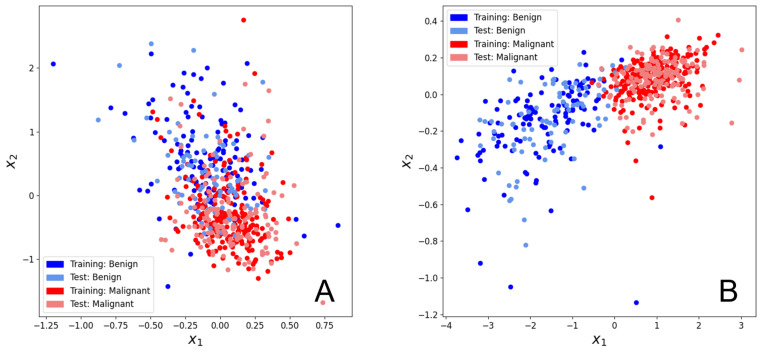
Scatter plots of the ($x_{1},x_{2}$) values of the breast cancer dataset with 30 starting features (corresponding to 60 trainable linear parameters). Datapoints from both the training set and the test set are depicted. A random seed of ’1’ was used for the train-test split of this data. The PennyLane software package was used to optimize the parameters [40]. (**A**) Scatter plot of the ($x_{1},x_{2}$) values after 0 training steps. (**B**) Scatter plot of the ($x_{1},x_{2}$) values after 1500 training steps.

**Figure 4 biomolecules-12-01576-f004:**
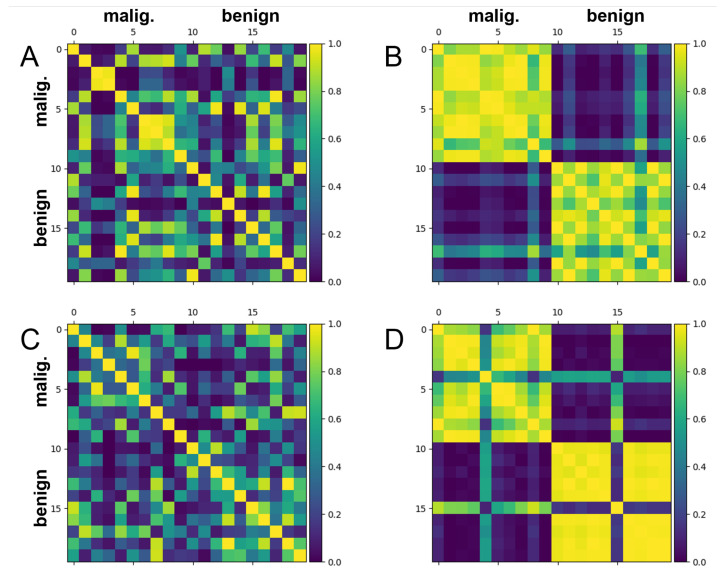
Gram matrices depicting mutual data overlap in Hilbert space (i.e., ${\left|\langle x|x^{{}^{\prime }}\rangle \right|}^{2}$) for 10 benign and malignant train and test samples from the breast cancer dataset. In each case, 30 starting features (corresponding to 60 trainable linear parameters) were used, with no subsequent PCA feature reduction. The stronger the separation between the purple tiles (benign) and the yellow tiles (malignant), the better the model’s ability to classify. A random seed of ’1’ was used for the train-test split of this data. The PennyLane software package was used to train the embedding [40]. (**A**) Mutual data overlap in Hilbert space for training set datapoints at optimization step 0. (**B**) Mutual data overlap in Hilbert space for training set datapoints at optimization step 1500. (**C**) Mutual data overlap in Hilbert space for test set datapoints at optimization step 0. (**D**) Mutual data overlap in Hilbert space for test set datapoints at optimization step 1500.

**Figure 5 biomolecules-12-01576-f005:**
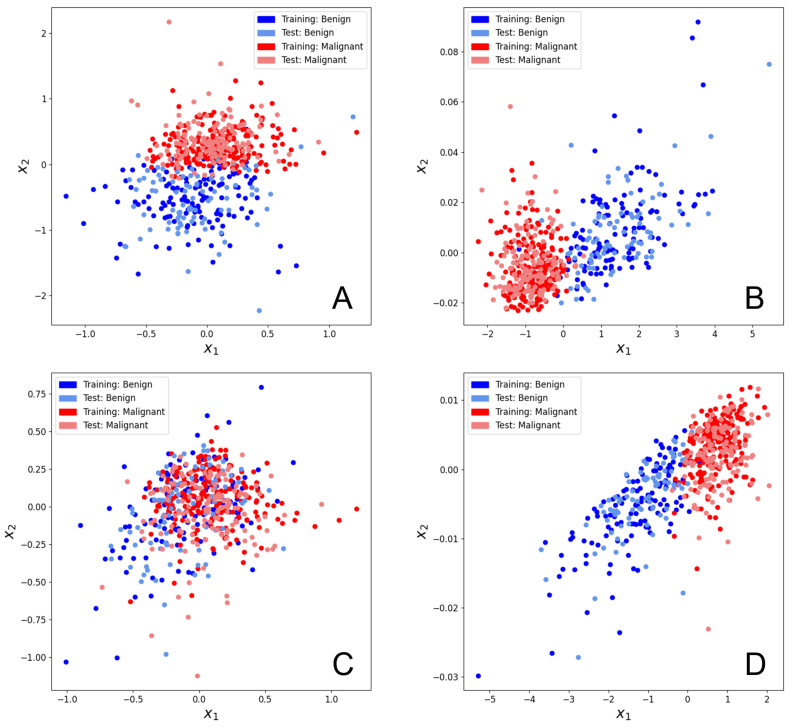
Scatter plots of the ($x_{1},x_{2}$) values of the breast cancer dataset following feature reduction through PCA. Datapoints from both the training set and the test set are depicted. A random seed of ’1’ was used for the train-test split of this data. The PennyLane software package was used to optimize the parameters [40]. (**A**) Scatter plot of ($x_{1},x_{2}$) values associated with 8 principal components after 0 training steps. These 8 principle components correspond to 16 trainable linear parameters. (**B**) Scatter plot of ($x_{1},x_{2}$) values associated with 8 principal components after 1500 training steps. These 8 principle components correspond to 16 trainable linear parameters. (**C**) Scatter plot of ($x_{1},x_{2}$) values associated with 4 principal components after 0 training steps. These 4 principle components correspond to 8 trainable linear parameters. (**D**) Scatter plot of ($x_{1},x_{2}$) values associated with 4 principal components after 1500 training steps. These 4 principle components correspond to 8 trainable linear parameters.

**Figure 6 biomolecules-12-01576-f006:**
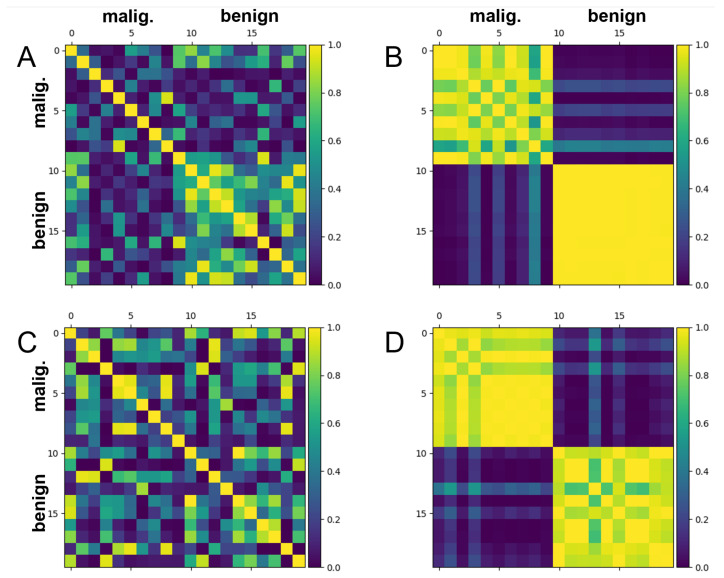
Gram matrices depicting mutual data overlap in Hilbert space (i.e., ${\left|\langle x|x^{{}^{\prime }}\rangle \right|}^{2}$) for 10 benign and 10 malignant train and test samples from the breast cancer dataset. In each case, PCA was used to reduce the number of features (and thus the number of trainable linear parameters). The stronger the separation between the purple tiles (benign) and the yellow tiles (malignant), the better the model’s ability to classify. A random seed of ’1’ was used for the train-test split of this data. The PennyLane software package was used to train the embeddings [40]. (**A**) Mutual data overlap in Hilbert space for test set datapoints at optimization step 0, using ($x_{1},x_{2}$) values generated from 8 principal components. (**B**) Mutual data overlap in Hilbert space for test set datapoints at optimization step 1500, using ($x_{1},x_{2}$) values generated from 8 principal components. (**C**) Mutual data overlap in Hilbert space for test set datapoints at optimization step 0, using ($x_{1},x_{2}$) values generated from 4 principal components. (**D**) Mutual data overlap in Hilbert space for test set datapoints at optimization step 1500, using ($x_{1},x_{2}$) values generated from 4 principal components.

**Table 1 biomolecules-12-01576-t001:** Test set assessment outcomes for training performed on the Hymenoptera dataset’s training set. Corresponding training costs are also given. In each row, training was performed for 1500 iterations using the root mean squared propagation optimizer (step size of 0.01) and a batch size of 10. All values are given to four decimal places. The features in row 1 did not undergo PCA, while the features from the rest of the rows did. A random seed of ’123’ used for the train-test split in every row other than the first (the first row used the default train-test split of the Hymenoptera dataset). The same random seed of ’123’ was used for all subsequent evaluations in all rows. The best value for each column is shown in bold.

No. of Features	ResNet (y/n)	Training Cost	Test Cost	Precision	Recall	F1-Score
512	y	**0.0141**	0.9931	**0.6184**	0.5663	**0.5912**
256	y	0.9944	0.9885	0.5326	**0.5976**	0.5632
256	n	0.9947	**0.9859**	0.4945	0.5488	0.5202
64	y	0.9756	0.9942	0.4891	0.5488	0.5172
64	n	0.9956	0.9928	0.4828	0.5122	0.4970
16	y	0.9926	0.9897	0.5000	0.5488	0.5233
16	n	0.9969	0.9892	0.4831	0.5244	0.5029
4	y	0.9909	0.9911	0.4545	0.4878	0.4706
4	n	0.9959	0.9947	0.4783	0.5366	0.5057
2	y	0.9700	0.9928	0.4545	0.4878	0.4706
2	n	0.9954	0.9965	0.4316	0.5000	0.4633

**Table 2 biomolecules-12-01576-t002:** Test set assessment outcomes for training performed on the UCI ML Breast Cancer Wisconsin (Diagnostic) Dataset training set. Corresponding training costs are also given. In each row, training was performed for 1500 iterations using the root mean squared propagation optimizer (step size of 0.01) and a batch size of 10. All values are given to four decimal places. The features in row 1 did not undergo PCA, while the features from the rest of the rows did. A random seed of ’123’ was used in each row, for both the train-test split and for all subsequent evaluations. The best value for each column is shown in bold.

No. of Features	Training Cost	Test Cost	Precision	Recall	F1-Score
30	0.1727	0.3623	0.9032	0.9790	0.9396
30	**0.1465**	0.3751	0.9091	0.9790	0.9428
16	0.2692	0.3023	**0.9338**	0.9860	0.9592
8	0.2757	**0.2903**	0.9226	**1.0000**	**0.9597**
4	0.2569	0.3440	0.9156	0.9860	0.9495
2	0.3953	0.3817	0.8981	0.9860	0.9400

**Table 3 biomolecules-12-01576-t003:** Test set assessment outcomes for training performed on the UCI ML Breast Cancer Wisconsin (Diagnostic) Dataset training set. Corresponding training costs are also given. In each row, training was performed for 1500 iterations using the root mean squared propagation optimizer (step size of 0.01) and a batch size of 10. All values are given to four decimal places. The features in row 1 did not undergo PCA, while the features from the rest of the rows did. A random seed of ’1’ was used in each row, both for the train-test split and for all subsequent evaluations. The best value for each column is shown in bold.

No. of Features	Training Cost	Test Cost	Precision	Recall	F1-Score
30	0.2026	0.2791	0.9205	0.9720	0.9456
30	**0.1750**	0.2899	0.9211	0.9790	0.9492
16	0.2201	0.3101	0.9281	**0.9930**	0.9595
8	0.2497	**0.2646**	**0.9655**	0.9790	**0.9722**
4	0.2885	0.2913	0.9467	**0.9930**	0.9693
2	0.3450	0.3306	0.9517	0.9650	0.9583

## Data Availability

The ImageNet Hymenoptera dataset can be accessed on Kaggle: https://www.kaggle.com/datasets/melodytsekeni/hymenoptera-data, accessed on 21 October 2022. The Breast Cancer Wisconsin (Diagnostic) Data Set can be accessed through UCI: https://archive.ics.uci.edu/ml/datasets/breast+cancer+wisconsin+(diagnostic), accessed on 21 October 2022. The code used in this manuscript can be accessed on GitHub: https://github.com/Rlag1998/QML_Generalization, accessed on 21 October 2022.

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
