# Peer review of "Generalization Performance of Quantum Metric Learning Classifiers"

_biomolecules, 2022, doi:10.3390/biom12111576_

Round 1
Reviewer 1 Report
The authors applied a kernel-based machine-learning algorithm termed quantum metric learning to separate training samples, ideally on a Hymenoptera dataset containing images of ants and bees. They also examined the method in the breast cancer dataset consisting of a significantly smaller number of features per sample while having a more significant sample number. The successful application of this approach requires a good balance between the initial number of sample features and the sample number of the dataset. The results are significant and contribute an essential application of machine learning to the field of quantum computing and simulations. The paper is well-written and presented in a straightforward style. I recommend its publication in Biomolecules.
Reviewer 2 Report
See Attachment
